# Spatial and temporal changes of SO₂ regimes over China in recent decade and the driving mechanism

Ting Wang[1], Pucai Wang[1, 3], Nicolas Theys[2], Dan Tong[4], François Hendrick[2],

Qiang Zhang[4], Michel Van Roozendael[2]

CAS Key Laboratory of Middle Atmosphere and Global Environment Observation,

Institute of Atmospheric Physics, Chinese Academy of Sciences, Beijing, China

Belgian Institute for Space Aeronomy (IASB-BIRA), Brussels, Belgium
University of Chinese Academy of Sciences, Beijing 100049, China
Ministry of Education Key Laboratory for Earth System Modeling, Department of

Earth System Science, Tsinghua University, Beijing, China

Revised, *Atmos Chem Phys*

Correspondence to:

Ting Wang, Institute of Atmospheric Physics, Chinese Academy of Sciences, Beijing

100029, E-mail: wangting@mail.iap.ac.cn

**Abstract:** The spatial and temporal changes of $SO_2$ regimes over China during 2005 to 2016 and their associated driving mechanism are investigated based on a state-of-the-art retrieval dataset. Climatological $SO_2$ exhibits pronounced seasonal and regional variations, with higher loadings in wintertime and two prominent maxima centered in the North China Plain and the Cheng-Yu District. In the last decade, overall $SO_2$ decreasing trends have been reported nationwide, with spatially varying downward rates according to a general rule—the higher the $SO_2$ loading, the more significant the decrease. However, such decline is in fact not monotonic, but instead four distinct temporal regimes can be identified by empirical orthogonal function analysis. After an initial rise at the beginning, $SO_2$ in China undergoes two sharp drops in the periods 2007-2008 and 2014-2016, amid which 5-year moderate rebounding is sustained. Despite spatial coherent behaviors, different mechanisms are tied to North China and South China. In North China, the same four regimes are detected in the time series of emission that is expected to drive the regime of atmospheric $SO_2$, with a percentage of explained variance amounting to 81%. Out of total emission, those from industrial sector dominate $SO_2$ variation throughout the whole period, while the role of household emission remains uncertain. In contrast to North China, $SO_2$ emissions in South China exhibit a continuous descending tendency, due to the coordinated cuts of industrial and household emissions. As a result, the role of emissions only makes up about 45% of the $SO_2$ variation, primarily owing to the decoupled pathways of emission and atmospheric content during 2009 to 2013 when the emissions continue to decline but atmospheric

content witnesses a rebound. Unfavorable meteorological conditions, including
deficient precipitation, weaker wind speed and increased static stability, outweigh the
effect of decreasing emissions and thus give rise to the rebound of $SO_2$ during 2009 to

46    2013.

**Key words:** $SO_2$, China, spatiotemporal regimes, mechanism, emission inventory,
meteorological condition



## 1 Introduction

In recent decade, air pollution has persistently plagued China, especially in leading economic and densely populated areas (Chan and Yao 2008; Ma et al., 2012; Chai et al., 2014). In China, environmental protection agencies identify six major pollutants of concern, including sulfur dioxide ($SO_2$), nitrogen dioxide ($NO_2$), ozone ($O_3$), carbon monoxide (CO), fine particulate matter (PM2.5) and coarse particulate matter (PM10). Then, values of the six pollutants are transformed into a single number called Air Quality Index (AQI) for effective communication of air quality status and corresponding health impact (MEPC, 2012).

$SO_2$ is one of the six major pollutants in China (Ren et al., 2017). It is harmful to human health, affecting lung function, worsening asthma attacks and aggravating existing heart disease (WHO, 2018). It also leads to the acidification of the atmosphere, and the formed sulfate aerosol is one of the most important components of fine particles in cities (Meng et al., 2009). Overall, $SO_2$ is a key influencing factor for atmospheric pollution, and it poses great threats to life, property and environment (Wang et al., 2014).

Compared to airborne and ground-based remote sensing, satellite platforms permit near-global coverage on a continuing and repetitive basis, enabling quick and large-scale estimation of pollution patterns (Yu et al., 2010). Since the world's first weather satellite TIROS-I launched in 1960, satellites have become a crucial part of Earth's observations and practical applications (Yu et al., 2010). Till now, $SO_2$ has been measured globally by several operational satellite instruments, such as OMPS (Zhang

et al., 2017), GOME-2 (Munro et al., 2006; Rix et al., 2012) and OMI (Lee et al., 2011;
Li et al., 2013; Theys et al., 2015).
With the aid of satellite data, in the past decade, various attempts have been made to
explore the variation of $SO_2$ loadings in China. Lu et al. (2010) report that total $SO_2$
emissions in China have increased by 53% from 2000 to 2006, followed by a growth
rate slowdown and the start of a decrease. Li et al. (2010), Yan et al. (2014), and Zhang
et al. (2012) all highlight the prominent reduction of $SO_2$ during 2007 and 2008, as a
consequence of the widespread deployment of flue-gas desulfurization and the strict
control strategy implemented for preparation of the 2008 Olympic Games. Throughout
the past decade, 90% of the locations in China have shown a decline in $SO_2$ emissions,
as highlighted by Koukouli et al. (2016). Such widespread declines are ascribed to
effective air quality regulations enforced in China (van der A et al., 2017). Furthermore,
Krotkov et al. (2016) and Li et al. (2017) both compared the sulfurous pollution in
China and India, and pointed out their opposite trajectories. Since 2007, emissions in
China have declined while those in India have increased substantially. Nowadays, India
is overtaking China as the world's largest emitter of anthropogenic $SO_2$. In addition,
several studies conducted analyses on $SO_2$ in sub-regions of China, for example Jin et
al. (2016), Lin et al. (2012), Wang et al. (2015) and Su et al. (2011). All these studies
contributed to a better understanding of $SO_2$ changes in China. However, there are still
key issues to be addressed. First, with the pace of considerable progress made on $SO_2$
retrieval, updated data products are now available to accurately derive recent $SO_2$
variations in China. Second, although the general decreasing tendency has been
revealed, the specific spatial and temporal regimes remain unclear. Does the $SO_2$
decrease monotonically, or is there a complicated oscillation? How similar/different are
$SO_2$ variations in different parts of the country? Third, there is more to be learned about
the driving mechanisms that govern $SO_2$ variations. Previous studies have mainly
focused on the impact from amounts of emission. However, the $SO_2$ content is not only
dependent on emissions but also on atmospheric conditions. Therefore, how large is the
influence of atmospheric variability on the variation of $SO_2$?
The overall goal of this study is to quantify the spatial and temporal changes of $SO_2$
regimes over China in the last decade and to disclose the driving mechanism, based on
a new-generation of $SO_2$ retrieval dataset (Theys et al., 2015). Figure S1 labels the
provinces of China. The manuscript is organized as follows. Section 2 describes the
new $SO_2$ product, and emission inventories and atmospheric data are introduced. In
Section 3, we evaluate the general patterns of $SO_2$ including mean distribution, long-
term trends and seasonality. Subsequently, Section 4 identifies the specific regimes of
$SO_2$ variability and the associated driving mechanisms. Finally, concluding remarks and
future directions are presented in Section 5 and Section 6, respectively.

**2   Data**
**2.1   $SO_2$ VCD retrievals**
The Ozone Monitoring Instrument (OMI) is one of four sensors onboard the Aura

satellite launched in July 2004 (Levelt, J. et al., 2006). In recent years, Belgian Institute

for Space Aeronomy (BIRA) and cooperators have developed an advanced Differential

Optical Absorption Spectroscopy (DOAS) scheme to improve the retrieval accuracy of

$SO_2$ in troposphere. A $SO_2$ vertical column product is generated based on the algorithm

applied to OMI-measured radiance spectra (Theys et al. 2015). The retrieval scheme is

a based on a DOAS approach, including three steps: (1) a spectral fit in the 312-326 nm

range (other fitting windows are used for volcanic scenarios but are not relevant for this

study), (2) a background correction for possible bias on retrieved $SO_2$ slant columns,

(3) a conversion into $SO_2$ vertical columns through radiative transfer air mass factors

calculation, accounting for the $SO_2$ profile shape (from the IMAGES chemistry

transport model), geometry, surface reflectance and clouds.

Compared to the BRD OMI NASA $SO_2$ product, the BIRA retrievals proved to be

better both in terms of noise level and accuracy. The BIRA product is also fully

characterized (errors, averaging kernels, etc.). The improved OMI PCA $SO_2$ product of

NASA show similar performance and long-term trends as the BIRA product. The BIRA

$SO_2$ product has been validated in China with long-term MAX-DOAS data (Theys et

al., 2015; Wang et al., 2017).

The dataset is made available on a 0.25° and 0.25° regular latitude-longitude grid

over the rectangular domain 70-140°E, 10-60°N, and covers the period of 2005 to 2016

at monthly interval. In addition, a cloud screening is applied to remove measurements

with a cloud fraction of more than 30%. Other details can be found in Theys et al. (2015).

Given that missing values are often presented in satellite-retrieved product due to the
limitations of retrieval algorithms under adverse environments, it is necessary to
evaluate the availability of monthly $SO_2$ data relative to the entire period. As mapped
in Figure S2, there appears to be a substantial fraction of data gaps in western and
northeastern China, especially in the winter half year. This can be attributed to snow
cover surfaces and high solar zenith angles, which invalidate the measurability. As a
result, it may be problematic when sampling western and northeastern China. In
contrast, the completeness across eastern parts of China is generally more than 80%
regardless of the season, sufficient for inferring the spatial and temporal structures. In
what follows, the analysis is mainly confined to the eastern China to avoid issues related
to missing data.
**2.2    Emission Inventory**
The $SO_2$ emissions at national and provincial level are collected from the China
Statistical Yearbook on Environment, which is compiled jointly by the National Bureau
of Statistics and Ministry of Environmental Protection. It is an annual statistics
publication, with industrial and household emissions listed separately. Currently, this
publicly available dataset spans the period from 2003 to 2015, covering 31 provinces
in China other than Taiwan, Hong Kong and Macau. Industrial emissions refer to the
volume of $SO_2$ emission from fuel burning and production processes in the premises of
enterprises for a given period, while household emissions are calculated on the basis of
consumption of coal by households and the sulphur content of coal. Notice that power
generation is incorporated into industrial sources and emissions from transportation
sources are not reported. This emission inventory released in the official yearbook
(OYB for short) has been cited or used in several works, i.e. Li et al. (2017), Yan et al.
(2017), Hou et al. (2018), and etc.
Since a credible emission inventory is the key foundation of this study, the Multi-
resolution Emission Inventory for China (MEIC) developed by Tsinghua University (Li
et al., 2017; Zheng et al., 2018) has been adopted to verify the OYB inventory as well
as to corroborate our findings. The MEIC is a bottom-up emission inventory model
including more than 700 anthropogenic sources and then aggregated into five sectors:
power, industry, residential, transportation and agriculture. Unlike the OYB estimate,
emissions from power plants in MEIC are considered to be a single sector and presented
separately. Here, we use province-level emissions from 2003 to 2015, together with the
monthly gridded emissions at $0.25º×0.25º$ horizon resolution for the years 2008, 2010,
2012, 2014 and 2016. To be in line with the OYB inventory, transportation and
agriculture sectors are excluded when calculate summed emission, and the power sector
is folded into industrial sector.
Figure 1 compares the OYB and MEIC emission inventories in terms of both national
and regional scales. In addition, the other two candidates on national annual totals
including REASv2 (Kurokawa et al., 2013) and Zhao (Xia et al., 2016) are overlaid.
Figure 1a shows that considerable differences exist with regards to the magnitude
among the four datasets and in particular OYB emissions are generally lower than those
deduced from other inventories. However, their temporal variations are characterized
in a very similar manner. As further illustrated in the scatter plot of OYB against the
other three (Figure 1b), highly linear clustered markers with correlations above 0.92
confirm such temporal consistency. On even smaller regional scale, as shown in Figure
1c, high degrees of correspondence between OYB and MEIC overwhelm the whole
eastern China, with most correlations exceeding the 0.05 significant level. In
comparison, the western China features relatively less agreement, but it is not a major
concern in this study.
In short, all the datasets capture coherent temporal behaviors, despite the spread in
their magnitudes. We emphasize that this study is centered on the fluctuation patterns
rather than the magnitude itself. Therefore, the above evidences justify the use of the
OYB dataset in the following text. Meanwhile, in order to test whether results were
robust to using a different data set, all analyses have been repeated using the MEIC
inventory.
**2.3   Meteorological Fields**
The large-scale meteorological conditions are taken from Japanese 55-year
Reanalysis (JRA-55) data, prepared by the Japan Meteorological Agency (Kobayashi
et al., 2015; Harada et al., 2016). The variables analyzed include total column
precipitable water, horizontal wind and temperature at pressure levels.

## 3 General patterns of SO$_2$ over China

### 3.1 Mean distribution

Based on 12 years of SO$_2$ column data over China, Figure 2a shows the spatial pattern of long-term mean. Overall, SO$_2$ distribution is of great inhomogeneity in China, with two maximum centers: one is the North China Plain (NCP for short), and the other is Cheng-Yu (CY) district in Southwest China. In particular, SO$_2$ amount in NCP exceeds 1.2 DU. There are two essential causes responsible for high SO$_2$ loading in the two areas. On the one hand, combined effect of rapid economic and industrial development as well as population growth leads to a high degree of anthropogenic SO$_2$ emission. Figure 2c and Figure 2d show the emission strengths, defined as emitted SO$_2$ per unit area, in each province based on OYB and MEIC respectively. Note that in the rest of this paper, the terms "emission" or "emission amount" always refers to "per unit area emission". It is obvious that the two regions release above 8.0 tons/km$^2$ SO$_2$ per year, which is three times greater than the average level of China. Although OYB exhibits smaller magnitude of emissions than MEIC, the spatial patterns in terms of relative difference across space are generally consistent. On the other hand, as shown in Figure 2b, either of the two regions is surrounded or partly surrounded by mountains, which makes it difficult for the pollutants to dissipate.

In contrast, over the sparsely populated western part of China, low SO$_2$ concentrations of less than 0.2 DU are observed, except over some provincial capitals. Since western part of China is less affected by human activities, anthropogenic sources

of $SO_2$ are much smaller than natural emissions including emissions from terrestrial
ecosystems and oxidation of $H_2S$ to $SO_2$ (Wang et al., 1999). Between latitude 30-40°N,
for example, the $SO_2$ amount over the eastern regions (110-120°E) are 6-12 times
greater than western regions (80-110°E). In addition, note that the low-level $SO_2$
columns in western China are subject to large uncertainties and the background
correction is an important source of error. However, the western China with weak $SO_2$
signals/background $SO_2$ is not the subject of the present work, since we mainly focus
on the highly polluted eastern China.
Besides the NCP and CY regions with highest SO2 loadings, this study is also
interested in Yangtze River Delta (YRD) and Pearl River Delta (PRD), the other two
economic mega-urban zones in China. These four identified hotspots NCP, CY, YRD
and PRD are outlined in Figure 2a and will be specially examined in the following
discussion.
**3.2   Seasonal Cycle**
The annual total is decomposed into seasonal cycle, as shown in Figure 3. In eastern
China, about 35% of the annual totals is from winter, while $SO_2$ in summer only
accounts for 15%; the remaining 50 percent is almost equally divided in spring and
autumn. Seasonal variations measured in the fractional contribution are similar within
eastern China.
To unveil the underlying mechanism, Figure 4 illustrates the annual cycle of $SO_2$
VCDs in relation to sulphur emission, precipitable water and temperature at the four
hotspots. Intensive heating during winter in North China raises sulphur release.
However, emissions alone are not sufficient to explain the pronounced seasonality of
$SO_2$. The remaining variation is associated with the seasonal change of the
meteorological conditions. Temperature and humidity are cold and dry in winter due to
the influence of winter monsoon, which jointly weaken the rate of oxidation and wet
deposition. Thus, one expects that $SO_2$ molecules will have a longer lifetime and
therefore will accumulate easier. The opposite is true for summer, when chemical
reaction is active and wet removal is effective. In summary, both emission and
meteorological change explain the seasonality of the atmospheric $SO_2$ loadings.
Due to the climate transition from southern China to northern China, the annual range
of $SO_2$ rises progressively from south to north. NCP has the greatest amplitude of up to
1 DU, while there is virtually no annual cycle in PRD. Larger amplitude for $SO_2$ cycles
in NCP arises from the significantly reversed source-sink imbalance between summer
and winter. In contrast, the climate in PRD is characterized by smoother transition over
the whole year and there is no heating season, which explains the insignificant seasonal
variation of $SO_2$ in PRD. The other two regions CY and YRD have approximately the
same amplitude of 0.6 DU, because they are on the same line of latitude.
**3.3  Long-term trends**
Figure 5 depicts the spatial pattern of linear trends in annual and seasonal $SO_2$ from
2005 to 2016. Overall, apparent downward trends overwhelm most parts of eastern
China, while western China has experienced little change. In particular, the most
significant reduction occurred in the highly $SO_2$-polluted regions, with the decreasing
rates amounting to 0.1 DU/a. This result suggests that the governments and
communities in these economically developed regions have done its best to effectively
control environmental pollution, including energy saving, emission cut and adjustment
of energy consumption structure, shutdown of the most polluting factories, upgradation
of coal quality, etc. Besides, enforcement of environmental protection laws is becoming
more and more rigorous (van der A et al., 2017). Therefore, under collaborative efforts,
the $SO_2$ levels in these highly developed regions with high background concentration
have been decreasing markedly in the recent decade. Moreover, the pattern correlation
between mean (Figure 2a) and trends (Figure 5 top) of $SO_2$ reaches to –0.77, implying
that the downward rate over China can be summarized into a general rule—the higher
the $SO_2$ loading, the more significant the decrease.
Figure 5a-d portrays the long-term trends of $SO_2$ on seasonal basis. On the one hand,
every season has witnessed $SO_2$ reduction, with the strongest decrease occurring in
winter and autumn. Consequently, it can be concluded that the $SO_2$ decrease in winter
and autumn contribute most to the reduction of annual $SO_2$. On the other hand, the
highly $SO_2$-polluted regions have experienced the most pronounced decrease across all
seasons, which is consistent with the annual outcomes. It is noteworthy to point out that
a belt of large positive values extend along 40°N in winter (Figure 5a). This feature is
a known artefact related to the large solar zenith angles at high northern latitudes.
Last, we discuss the trends of the four hotspots interested. Figure 6 depicts the $SO_2$
columns from 2005 to 2016 as a function of year (y-axis) and calendar month (x-axis).
The horizontal axis is the month of the year, the vertical axis is the year, and the color
is the $SO_2$ VCD for that month and year. $SO_2$ VCDs exhibit a decreasing tendency
during the last decade, regardless of the time of the year. Quantitatively, $SO_2$ in NCP,
CY, YRD and PRD had undergone an overall downward trend with a rate of 0.062,
0.059, 0.046 and 0.055 DU per year, respectively.

**4   Specific regimes of $SO_2$ variability and causes**
**4.1   Specific regimes of $SO_2$ variability**
The above investigation presents $SO_2$ patterns and trends across China, but some
elusive non-monotonic behaviors are not fully understood. In this section, we aim to
detect the specific regimes of $SO_2$ variability and associated responsible mechanisms.
Spatiotemporal regimes of $SO_2$ over China are mapped by using empirical
orthogonal function (EOF) decomposition (Hannachi, 2004), which is a useful tool to
reduce the data dimensionality to two dimensions. One dimension represents the spatial
structure and the other the temporal dimension. Figure 7 illustrates the leading mode
(top) and the corresponding principal component (PC, bottom) obtained from EOF,
since only the first mode is statistically well separated. Compared to the first EOF mode
explaining 36.8% of the total variance, each of the other modes is characterized by less
than 6% contribution and thus discarded. On the one side, the variation of $SO_2$ is
dominated by a spatially uniform feature with large loadings in NCP and CY, suggesting
that $SO_2$ changes would be in the same phase but varying amplitude across the entire
region. On the other side, the corresponding PC exhibit overall declines during the 2005
to 2016. However, the result does not implicate a simple continuous decrease. In fact,
there appears to be a transient increase until a peak and thereafter two sharp drops occur
in the periods 2007-2008 and 2014-2016, amid which $SO_2$ concentrations are under the
process of slightly rebounding. In short, the $SO_2$ variability is characterized by four
distinct temporal regimes.
Moreover, Figure 8 demonstrates the time series for each province in eastern China,
with the segment over 2009-2013 highlighted by red color. It reflects extensive common
variation that goes through four stages—that is, a short-lived increasing period at the
beginning, a steep drop period during 2007 to 2008, a rebound period of 2009 to 2013
and another drastic drop period during 2014 to 2016. Most importantly, it confirms that
the $SO_2$ does not evolve in a monotonic way but shows a striking rebound during 2009
to 2013. This pattern is true throughout most of the region, with only two exceptions of
Guizhou and Guangdong provinces that had experienced a consecutive decrease since

318    2005.

**4.2   Causes**
In this section, we diagnose the likely mechanisms behind the observed $SO_2$
variability. Generally, emissions and meteorological conditions are two main factors
that essentially exert influence on atmospheric pollutant load. The impact of changes in
emitted $SO_2$, as the main driving force, is first examined. To this end, the temporal

classifications of $SO_2$ emission for each province based on OYB and MEIC are

respectively depicted in Figure 9a and 9b, in which red upward pointing triangle implies

non-monotonic decrease with a rebound in the middle whereas persistent decrease is

denoted by green downward pointing triangle. In North China except Henan province,

both OYB and MEIC datasets show that the emission passed its secondary peak during

2009 to 2013. In South China, however, discrepancies between OYB and MEIC emerge

in some provinces, namely Jiangxi, Hunan, Guangxi and Guizhou. Even so, we are still

confident enough that the majority of South China has witnessed a successive drop in

emitted $SO_2$. In addition, an auxiliary map is presented in Figure 9c showing the slope

of the linear regression of MEIC gridded emission over years 2008, 2010 and 2012. We

can see that most of North China is subject to a positive rate of change while the

opposite holds true over most of South China, which confirms the above findings.

Eventually, it comes to conclusion that despite spatially uniformity in temporal-pattern

classification of $SO_2$ VCD (Figure 8), temporal structure of emission demonstrates

strong south-north contrast (Figure 9). Therefore, it is advantageous to treat North

China and South China separately, as delineated by the dotted line in Figure 9. Regional

averaged quantities are estimated as a weighted average by assigning the district area

as a weight. In addition, to evade possible contaminations, we have ruled out Henan

and Jiangxi provinces in OYB and Henan, Hunan, Guizhou and Guangxi provinces in

MEIC.

Although we divide the eastern China into north and south blocks, the inter-regional

transport cannot be neglected. Therefore, we construct an Effective Emission Index
(*EEI*) to account for impacts from both local and remote sources. Here, we directly
adopt the results obtained by Zhang et al. (2015), who divided eastern China into three
parts North China, Southeast China and Southwest China, and quantified the percent
contributions of within-region versus inter-regional transport on sulfate concentrations.
The geographical partition in their work broadly coincides with ours, with the only
difference that South China is further split in two parts. Given that the ratio of Southeast
China to Southwest China is 1.4, we merge the percent contributions over these two
portions via simple conversions. This produces: for North China, within-region $SO_2$
emission contribute 68% followed by 19% from South China and 13% from other
regions; for South China, within-region emissions provide 66%, while transport from
North China and other regions amounts to 17% and 17% respectively. With these
statistics, the *EEI* is formulated as follows:
North China
$$EEI_1 = 0.68 + 0.19 + 0.13 = 1$$
$$EEI_m = 0.68 \cdot \frac{N_m}{N_1} + 0.19 \cdot \frac{S_m}{S_1} + 0.13$$

South China
$$EEI_1 = 0.17 + 0.66 + 0.17 = 1$$
$$EEI_m = 0.17 \cdot \frac{N_m}{N_1} + 0.66 \cdot \frac{S_m}{S_1} + 0.17$$

where *N* and *S* denote the emission amount in North China and South China respectively,
and subscripts 1 and *m* the 1st and *m*th time node respectively. The fundamental
assumptions to derive the formula are that *EEI* is linearly dependent on *N* and *S* and the
external contributions remain fixed (without interannual variation). For comparison
purpose, we also define an Emission Index (*EI*) that involves single effect from within-
in region emission, as written below,
North China
$$EI_1 = 1$$
$$EI_m = \frac{N_m}{N_1}$$

South China
$$EI_1 = 1$$
$$EI_m = \frac{S_m}{S_1}$$

where the notions of symbols are identical to those in *EEI* definition. In the case of
large scale, integrating the role of inter-regional transport does not alter the overall
pattern, as proved in the following analyses.
Figure 10 presents time series and scatter plots of $SO_2$ VCD and emission with its
variants *EI* and *EEI*, and Figure 11 is designed to show the total emission generated by
industries and households. These two figures are created based on OYB inventory,
while their counterparts obtained from MEIC inventory are shown in Figure S3 and S4
in the supplement material. As shown in Figure 10a, the North China features a good
correspondence between amount and either *EI* or *EEI*, with linear correlation of 0.9.
Time series of emission also indicate the existence of four distinct regimes that are
likely to drive the regime of $SO_2$ VCD directly. This is confirmed by the scatter plot
(Figure 10b), in which the points are tightly clustered around the regression line. Based
on variance analysis, emission accounts for 81% fraction of $SO_2$ VCD variation over
North China. In parallel, the same procedure relying on MEIC inventory yields nearly
identical results, as shown in Figure S3a and S3b. Furthermore, how large do industrial
and household sectors respectively contribute to the total trends? Figure 11a and Figure
S4a indicate that the industrial emissions play a crucial role in $SO_2$ VCD variation
throughout the whole period, while the influence induced by residential activity is
secondary. A more in-depth comparison between OYB and MEIC shows some
dissimilarity in household emission: OYB-based household emission acts to offset
industrial effect, while opposite function is identified for the MEIC-based one. However,
this does not seriously affect the major conclusion, due to the marginal impacts caused
by households.
The close linear relationship observed in North China is not found in South China,
since the two curves appear to become no adherent in Figure 10c and the points in the
scatter plot Figure 10d are widely spread around the regression line. Variance analysis
suggests that only 45% of $SO_2$ VCD variability is forced by emissions, suggesting that
the $SO_2$ variations in South China cannot be explained by emission changes alone. This
is mainly ascribed to the decoupled pathways of emission and $SO_2$ VCD during 2009
to 2013, as the emission continues to decline but $SO_2$ VCD witnesses a rebound. MEIC
emissions also exhibit a general decreasing tendency in spite of a transient pause
embedded, as shown in Figure S3c. Moreover, Figure 11b and Figure S4b suggests that
the cuts of industrial and household emissions collectively promote the continuous
decrease of total emission in South China, which are different from that in North China.
However, the emission decrease in the household sector is differently reported in the
OYB and MEIC inventories, the former one showing a sudden shift while the latter
displays a gradual decrease. Anyway, it is assured that household emissions in South
China have undergone a reduction, irrespective of the exact manner.
Why decreasing emissions do not cause a reduction of $SO_2$ VCD in South China
during 2009 to 2013? To answer this question, the atmospheric conditions during 2009
to 2013 are compared with those during the rest of the years, as depicted in Figure 12.
The period 2009 to 2013 is characterized by prolonged dry conditions in South China
with the precipitable water and precipitation being lower than usual (Figure 12a), which
weakens wet adsorption and scavenging. At the same time, this period is also associated
with relatively weaker wind speed (Figure 12b) and increased static stability (Figure
12c, d), reducing the ability of the atmosphere to diffuse leading to the accumulation of
$SO_2$ loads. In brief, unfavorable meteorological conditions produce the observed
rebound of $SO_2$ during 2009 to 2013, despite the continued decrease of emission.

## 5    Conclusions

In this study, the spatiotemporal variability of $SO_2$ columns over China and the
associated driving mechanisms are examined over the past decade. Based on a state-of-
the-art $SO_2$ retrieval dataset recently derived from the OMI instrument, we elaborate on
the characteristics of specific $SO_2$ regimes over China and underlying causes.
Climatological $SO_2$ in China has an uneven spatial distribution in space and time.
East China is far more exposed to $SO_2$ pollution than West China, with two maxima
centered in NCP and CY. From analysis of the annual cycles we conclude that 35% of
the annual totals are from winter, while $SO_2$ in summer only accounts for 15% percent.
In addition, the annual amplitude of $SO_2$ rises progressively from south to north.
From 2005 through 2016, most of eastern China presents a clear decreasing tendency
for $SO_2$, while western China has experienced little change. Spatially, the decreasing
rate is generally enhanced for high $SO_2$ loads. When computed seasonally, $SO_2$
reductions in winter and autumn contribute most to the reduction of annual $SO_2$.
Four stages of variation are identified by EOF analysis. The first regime (2005-2006)
features a transient increasing trend, the second (2007-2008) and the last (2014-2016)
regimes show sharp drops, and the third regime (2009-2013) manifests itself by 5-year
moderate rebounding. Although temporal regimes of $SO_2$ are coherent throughout the
country, different driving forces are tied to North China and South China. In North
China, the atmospheric $SO_2$ and emission varies essentially in the same way. Therefore,
the atmospheric $SO_2$ variability is primarily associated with the emission variability,
which accounts for 81% of the total variance. Further, the emission generated by
industrial sector is largely responsible for the atmospheric $SO_2$ variability. The
household emissions appears to remain uncertain, due to the dissimilarity between OYB
and MEIC inventories.
$SO_2$ emissions in South China exhibit a continuous decreasing tendency, due to the
coordinated cuts of industrial and household emissions. As a result, the role of
emissions only contributes 45% of the $SO_2$ variation, primarily owing to the decoupled
pathways of emission and atmospheric content during 2009 to 2013 when the emission
continues to decline but atmospheric content witnesses a rebound. It is found that such
rebound occurs in response to the joint effect of deficient precipitation, weaker wind
speed and increased static stability during 2009-2013.

## 6    Future directions

As enlightened by this study, the spatial and temporal changes of $SO_2$ regimes over
China in recent decade become clear. However, there is much left to be learned about
the responsible driving mechanisms. First, a major obstacle of cause-and-effect relation
surveys stems from uncertainties in the current emission inventories. In this study, many
facets inferred by OYB and MEIC are convergent, because we look at large spatial scale
and long-term general tendency that help filter out or attenuate some uncertainties.
However, if the aim is to focus on smaller spatial or temporal scales or on specific
sectors, there is still great uncertainty. To overcome these barriers, emission inventories
should be further improved and more observational products should be used for
comparison. Second, this work investigates the impact of emission, inter-regional
transport and meteorology using purely statistical techniques, but finer scale
investigations require numerical simulations using coupled chemical-transport models.
Third, the analysis presented in Section 4 is constrained to provincial or multi-
provincial levels, due to the limitation that only continuous emission data on provinces
are gathered at hand. In reality, however, either emission or atmospheric loadings can
be quite inhomogeneous within the same region. Therefore, future studies should use
both gridded $SO_2$ VCDs and gridded $SO_2$ emission inventories.


**Acknowledgement:** We are grateful to the editor and two anonymous reviewers for
constructive comments and suggestions that greatly improve quality of this paper. This
work was supported by the National Key Research and Development Program of China
nos. 2017YFB0504000 and 2016YFC0200403, and the National Natural Science
Foundation of China nos. 41505021 and 41575034.

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

596

597

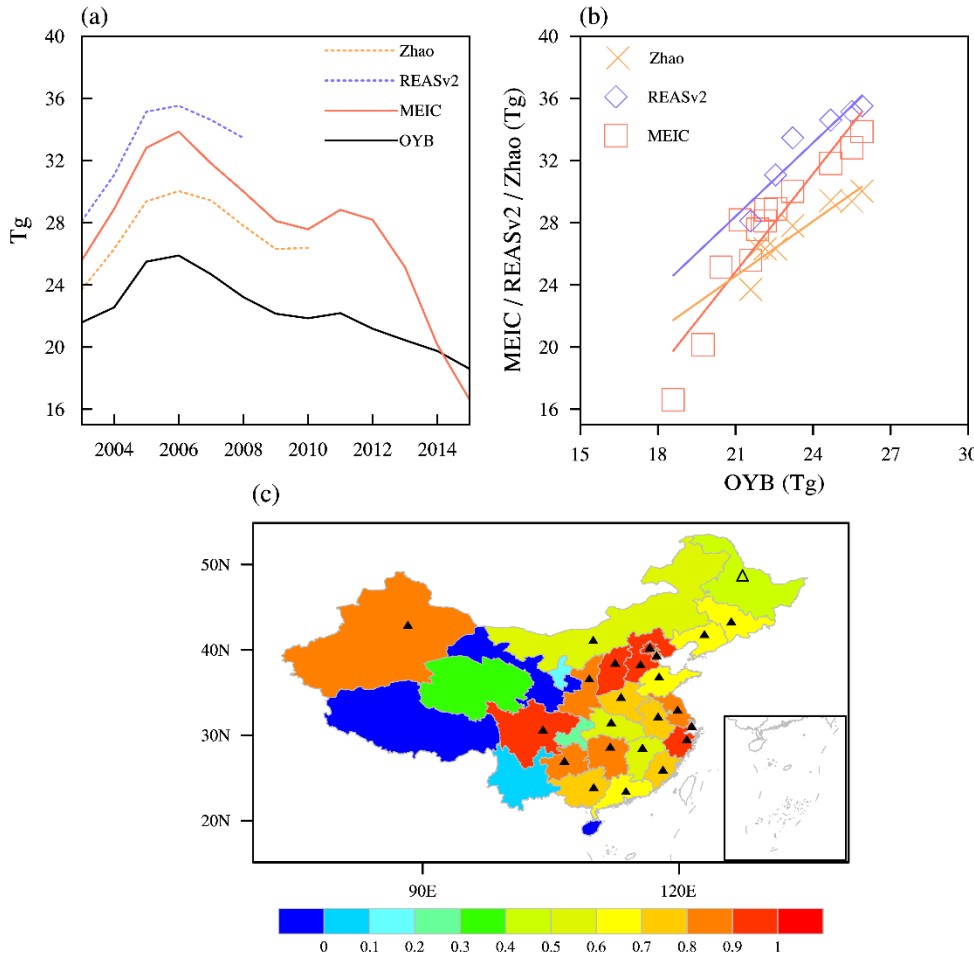

598

Figure 1 (a) National total SO$_2$ emissions estimated by OYB (solid black), MEIC (solid red), REAS

(dashed blue) and Zhao (dashed orange) between 2003 and 2015. (b) Scatter diagrams and regression

lines for OYB estimate (x-Axis) against the other three products (y-Axis). (c) The province-by-province

correlations between OYB and MEIC products, with the significance levels of 0.1 and 0.05 are marked

by open and filled triangles respectively.



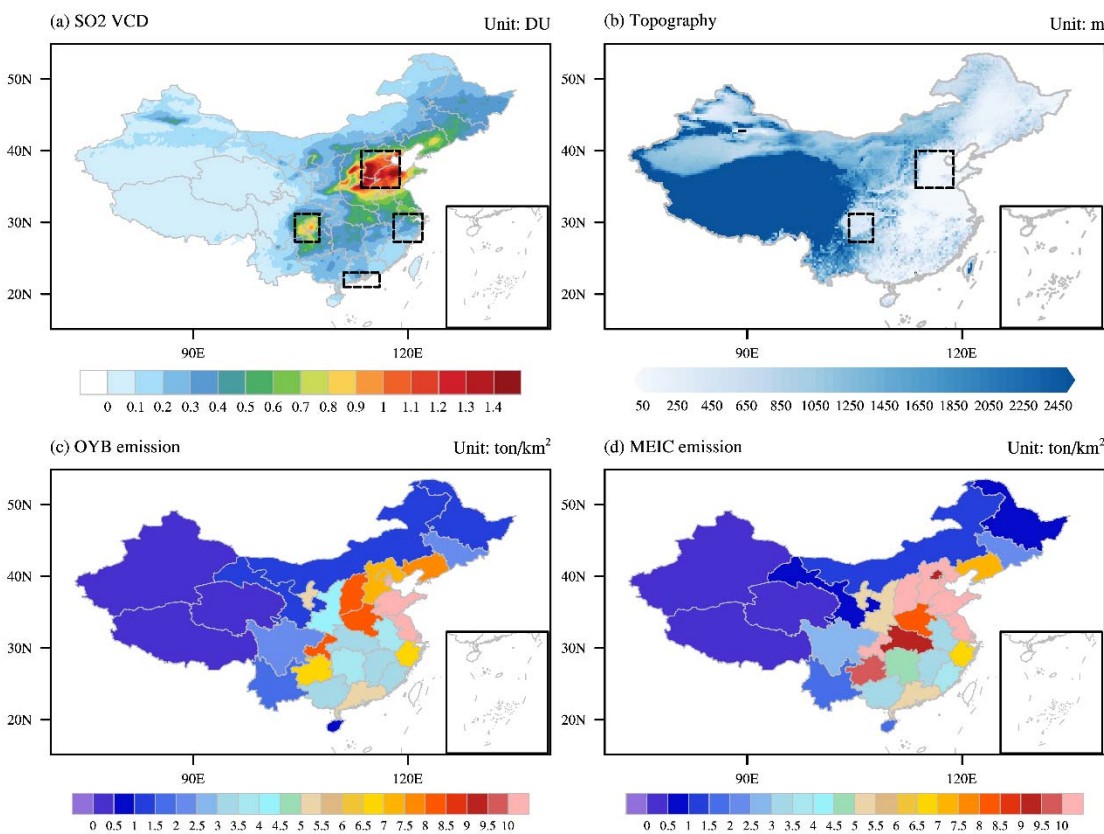


Figure 2    (a) Spatial distribution of 12-year (2005–16) averaged SO$_2$ columns over China. (b)
Topography of China in meters. (c, d) SO$_2$ emission (ton/km$^2$) among Chinese provinces based on OYB
and MEIC.


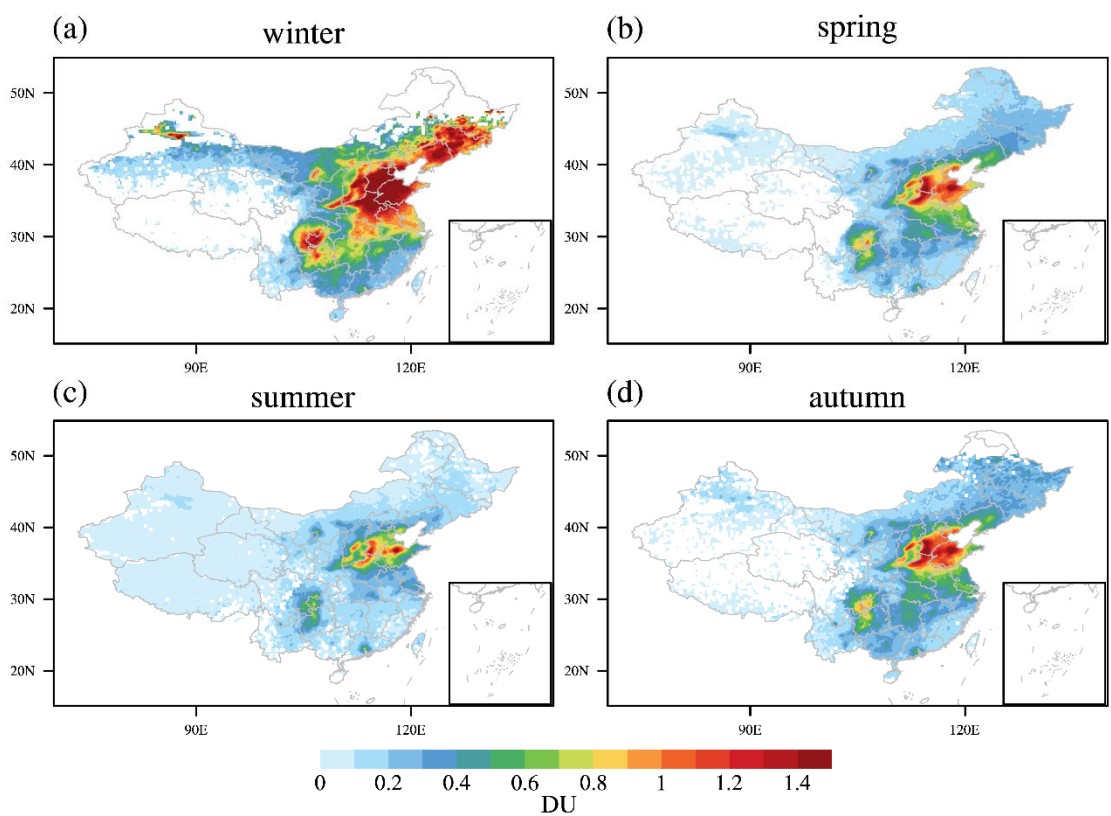


Figure 3 Seasonal SO$_2$ columns over China: (a) winter, (b) spring, (c) summer and (d) autumn





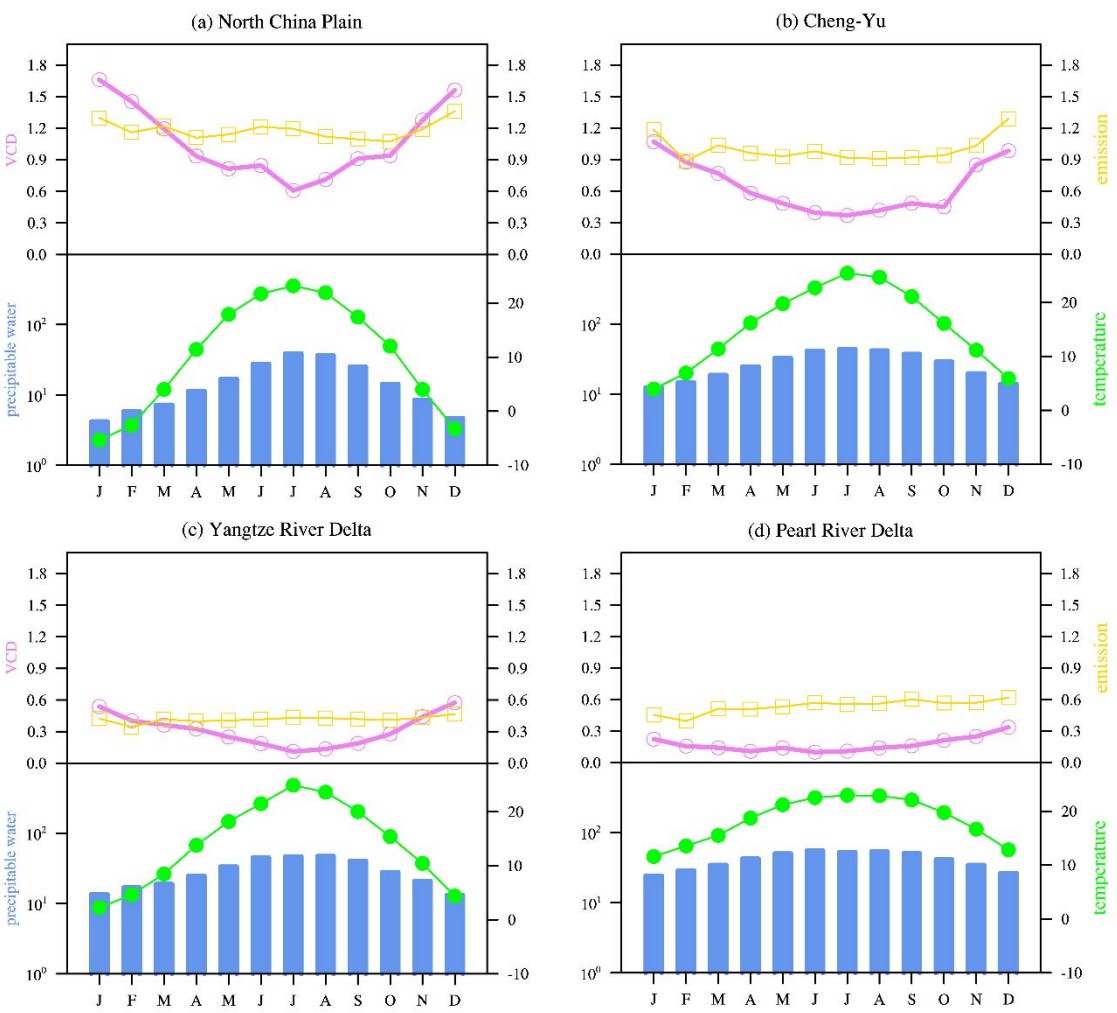


Figure 4 Annual cycle of SO$_2$ VCD (unit: DU, pink line), MEIC SO$_2$ emission (unit: ton/km$^2$, yellow

line), precipitable water (unit: kg/m$^2$, blue bar) and temperature at 925hPa (unit: ℃, green line) for NCP

(a), CY (b), YRD (c) and PRD (d).



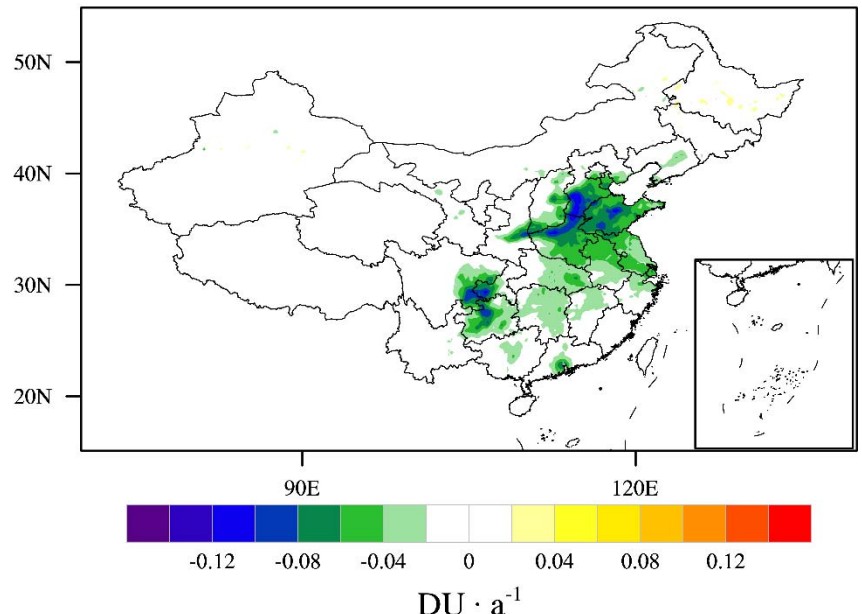


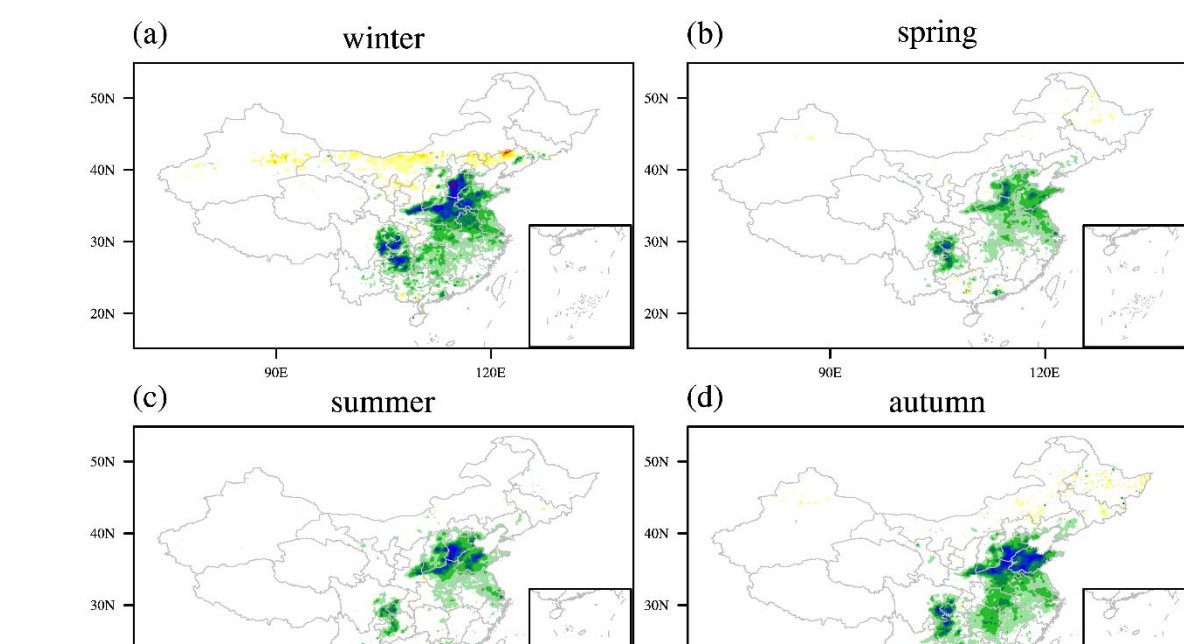


Figure 5    Spatial pattern of SO$_2$ linear trends (2005–16) in annual (Top) and seasonal values (a, b, c,

626                                         d)



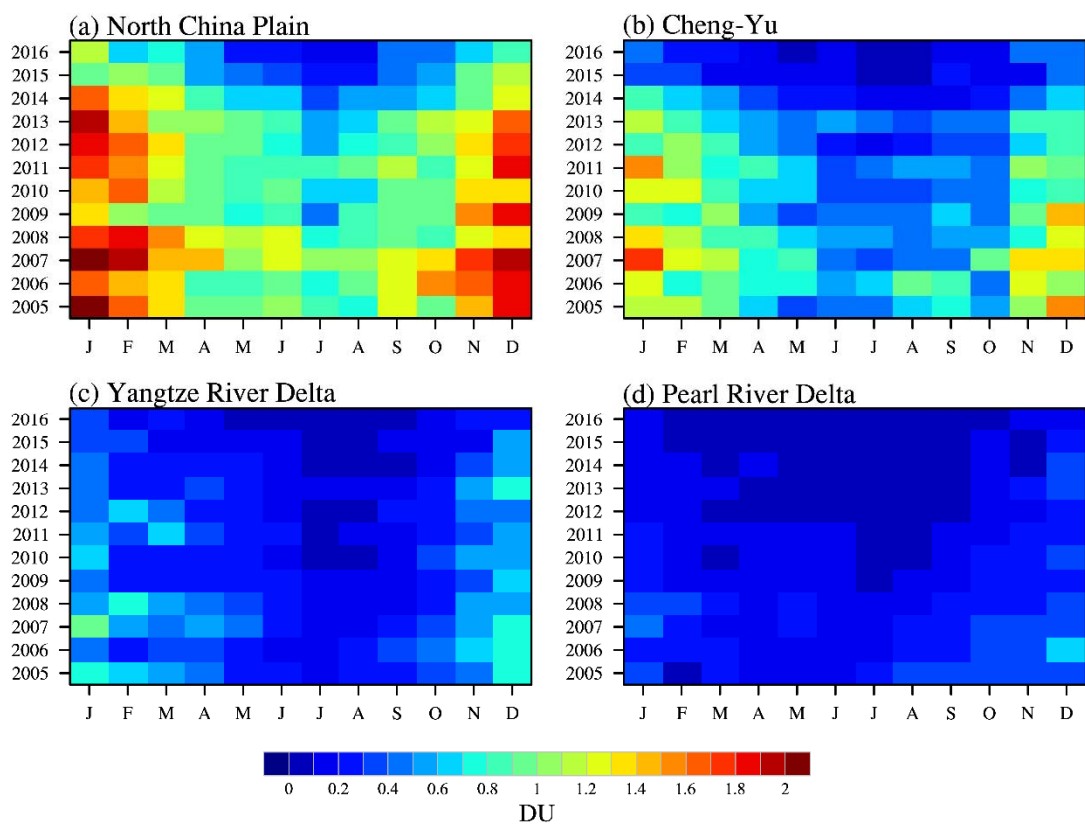


Figure 6    SO₂ amounts from 2005 to 2016 as a function of year (y-axis) and calendar month (x-axis) for
NCP (a), CY (b), YRD (c) and PRD (d).


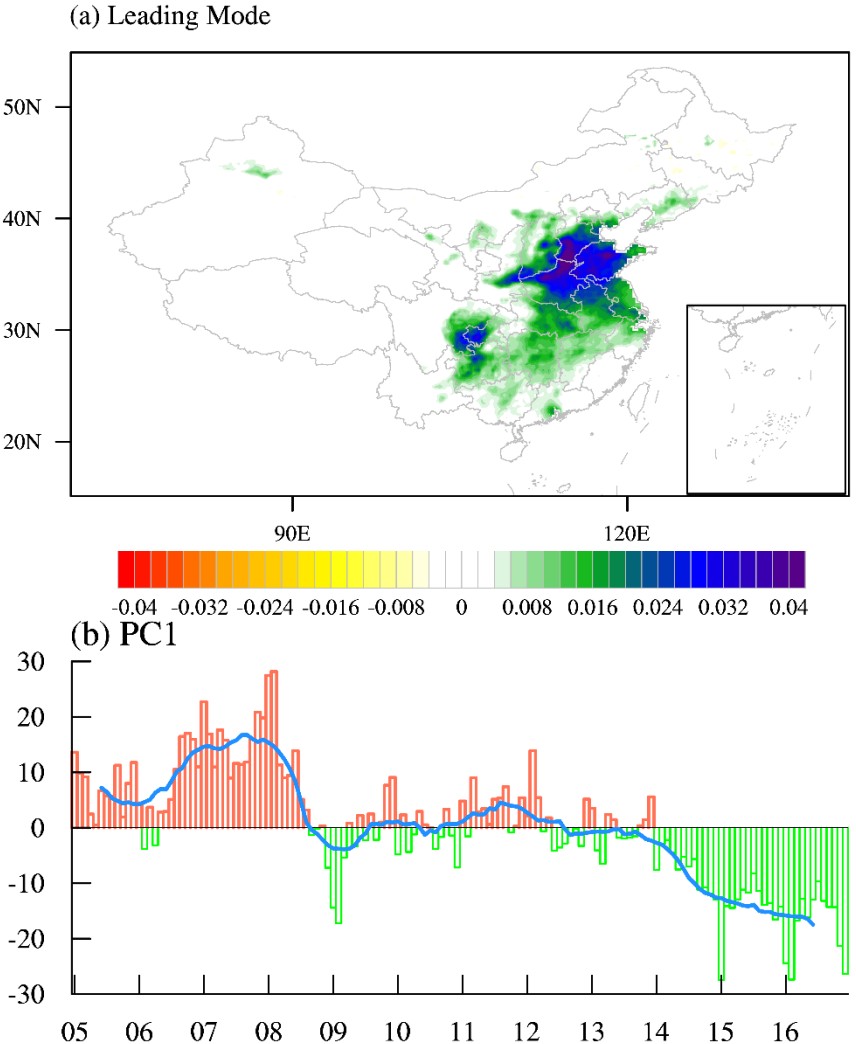


Figure 7    The first leading EOF mode (a) and the corresponding principal components (b)




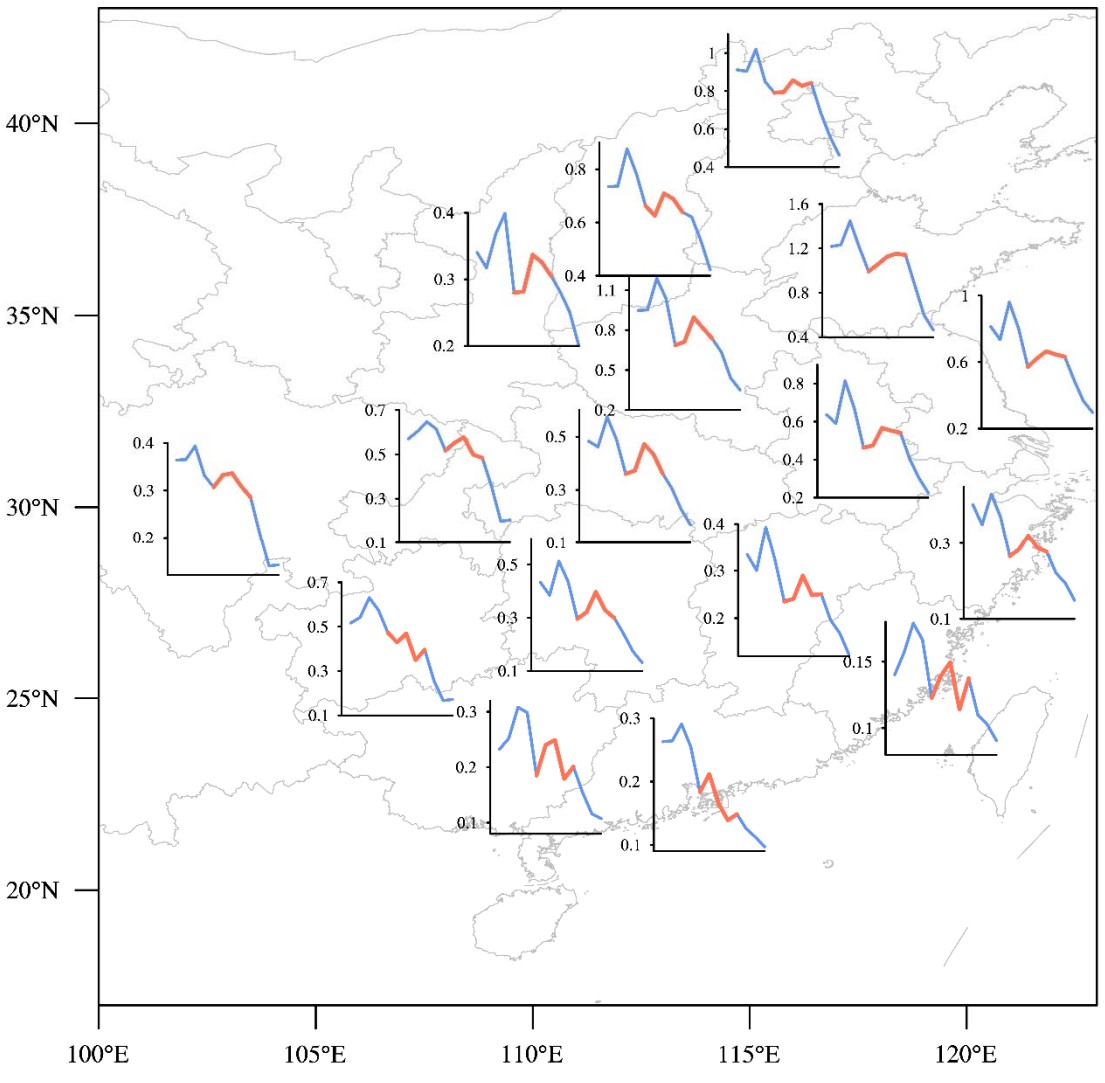


Figure 8 Temporal evolution of annual SO$_2$ (unit: DU) from 2005 to 2016 in each province of eastern

China, with the segment over 2009-2013 highlighted by red color.



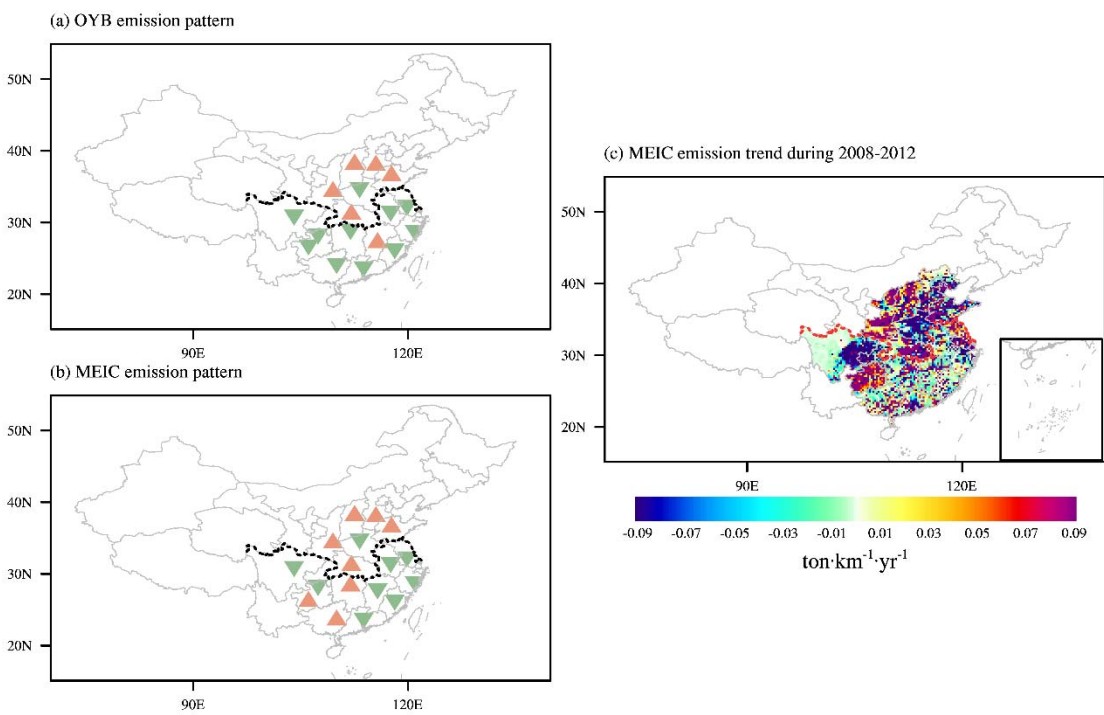


Figure 9    (left panel) Temporal structure classification of SO$_2$ emission based on OYB and MEIC. Red

upward pointing triangle implies non-monotonic decrease with a rebound in the middle, while monotonic

decrease is denoted by green downward pointing triangle. (Right panel) slope of the linear regression of

MEIC gridded emission over years 2008, 2010 and 2012. The black or red dotted line delimits the North

China and South China.


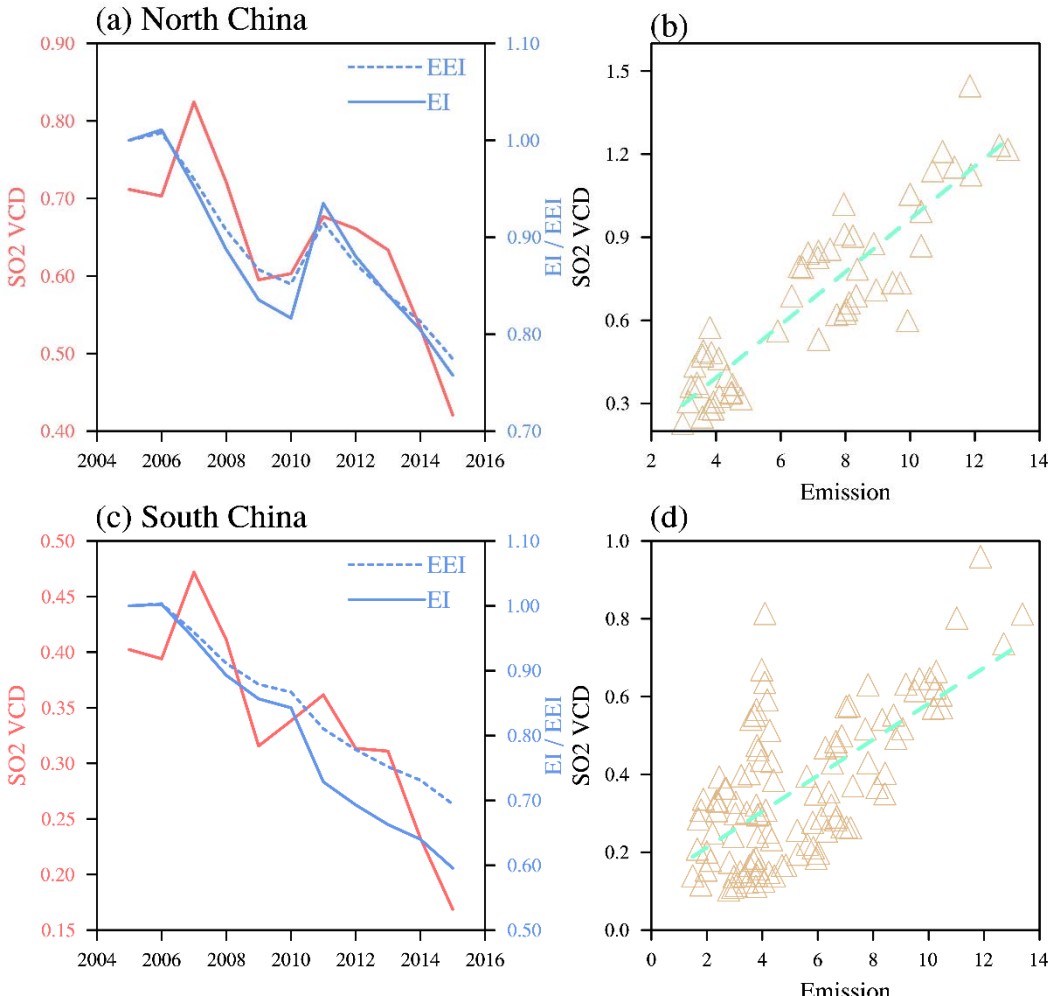


Figure 10    Time series plots of SO2 VCD and EI/EEI (a, c), and scatter plots with regression line of

SO2 VCD and emission (b, d) for North China (1st Row) and South China (2nd Row). Each marker in b

and d corresponds to one year and one province.



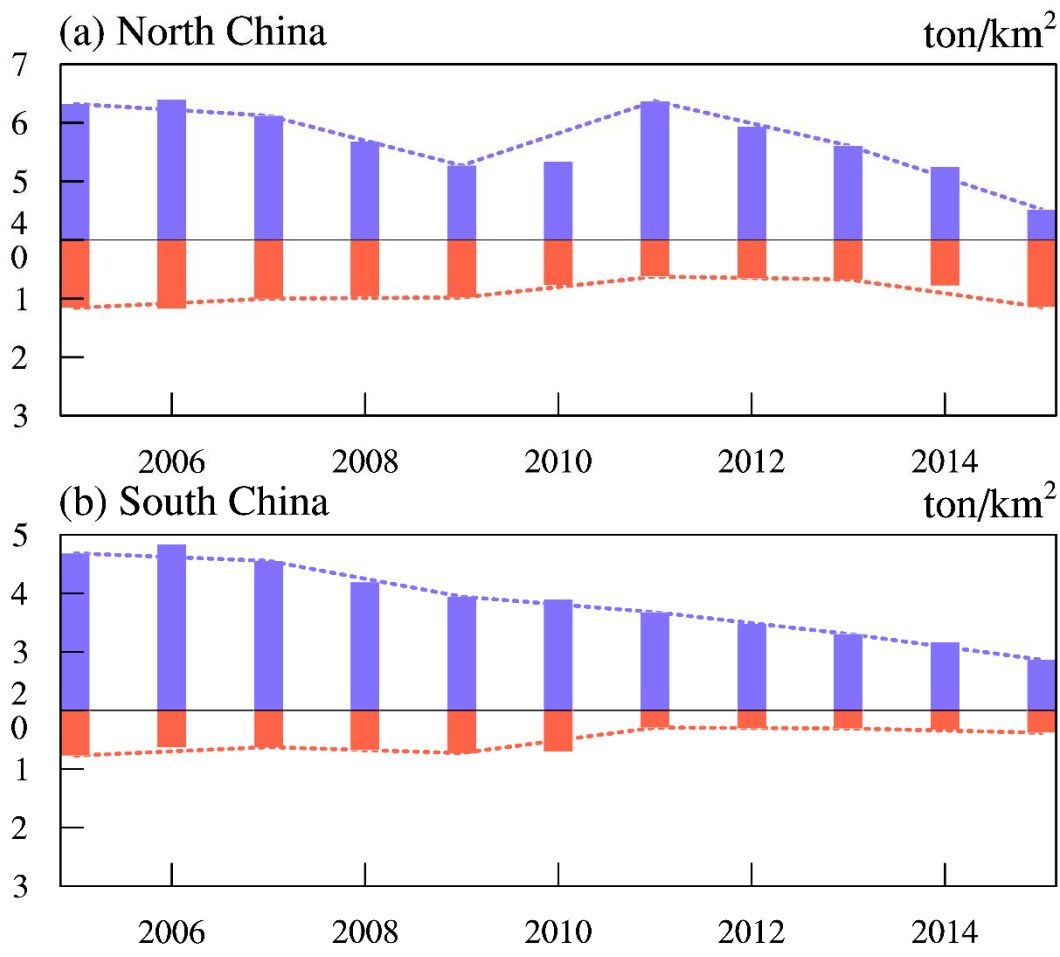


Figure 11    Annual SO2 emission (ton/km²) generated by industries (upward blue bars) and households
(downward red bars) in North China (a) and South China (b). Notice that the Y-axis in a positive direction
does not start at zero.


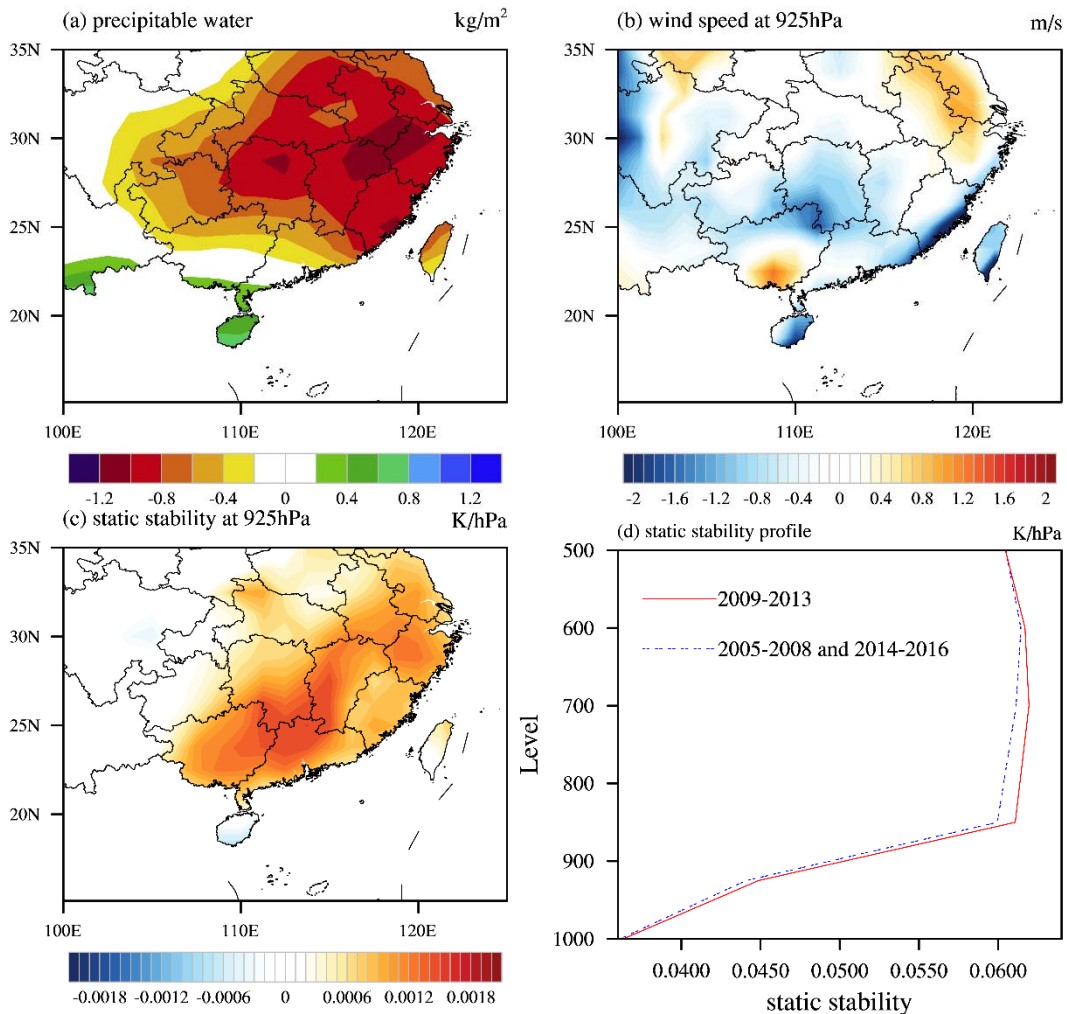


Figure 12 Comparison of atmospheric conditions between the period of 2009-2013 and the other years:

(a) composite difference in precipitable water (unit: kg/m$^2$), (b) composite difference in wind velocity at

925hPa (unit: m/s), (c) composite difference in static stability at 925hPa (unit: K/hPa), and (d) averaged

vertical profile of static stability over the 23-31N°, 105-122°N rectangle for the two episodes.



