# Peer review of "Spatial and temporal changes of SO2 regimes over China in"

_Atmospheric Chemistry and Physics, 2018_

## Referee Comment (RC1) · Anonymous Referee #1 · 17 Aug 2018

This paper uses OMI SO2 retrievals to study the effects of emissions and meteorology on SO2 loading over eastern China during 2005-2016. Monthly OMI SO2 from BIRA DOAS retrievals are compared with estimated SO2 emissions from the China Statistical Yearbook. The authors show that OMI-observed SO2 has decreased significantly over eastern China, particularly for areas with the strongest emissions. They use an EOF analysis to demonstrate that the change is not monotonic and has four phases, with SO2 increasing during 2005-2007, decreasing sharply during 2007-2008 and 2014-2016, and only slightly increasing or decreasing during 2008-2013. They also show that the emissions and OMI SO2 are highly correlated over northern part of the country, but less so for southern China. They propose that abnormally dry and

stagnant conditions over southern China during 2008-2013 may have caused a slight increase in SO2 loading, despite continued reduction in emissions. While several studies have examined the recent changes in SO2 pollution over China using satellite data, this study attempts to provide a somewhat different perspective. The conclusion that meteorology may play a fairly prominent role in the inter-annual changes in SO2 over southern China is interesting. The paper is well-organized and figures are mostly clear. However, I am not completely convinced that the emission data used can fully support the conclusions drawn in the study. I'd recommend that major changes be made before the paper can be accepted for publication in Atmos. Chem. Phys.

Specific comments: The authors indicate that the emission data used in this study have very strong seasonal changes in SO2 emissions from China (almost half of emissions in winter, and only 10% in summer). But this is quite different from many previously published emission inventories which generally suggest a much smaller seasonal change (such as HTAP). Also according to a number of previous studies, the residential sector is in general estimated to contribute roughly 10% of all SO2 emissions. This is quite different from what is shown in Figure 9 of this study. The authors may consider using a different, more widely recognized emission inventory for their analysis and check if their conclusion still stands.

It also appears that the emission data used here are on a provincial level (and not gridded) and the authors calculate the emission strength based on the area of each province. Can the authors confirm that? If so, how do the authors calculate total emissions (for example those in Figure 8) for a domain that partially covers several different provinces? Also note that the emissions and SO2 loading can be quite inhomogeneous even within the same province.

It is not clear how the "north" and "south" are defined in this study. One would assume that Cheng-Yu, PRD, and YRD are all part of the "South". But the SO2 time series in Figure 6 indicates that they have different trends during 2008-2013. How would the authors explain these different trends when Figure 10 appears to show generally

similar meteorological conditions for the three regions?

The "sudden downward shift of household emissions" in the south is quite surprising. Do the authors have an explanation for this? Or is this simply indicative of methodology change in the emission inventory?

Figure 4: there seem to be some negative SO2 values in the figure? Can the authors confirm that?

Figure 8: What is the unit for emissions? What does each data point represent in the scatter plot?

Figure 10: which area is the vertical profile in (d) for?

Writing: the authors should also make an attempt to improve the writing. Short, simple sentences in some cases may make the paper easier to follow.

―――――――――――――――――――

---

## Referee Comment (RC2) · Anonymous Referee #2 · 27 Aug 2018

General comments: In this paper, Ting Wang et al. analyzed the spatial distribution and temporal variation trends of SO2 VCD and emissions in different regions of China in the last decade based on the OMI observation and emission inventory. Further they discussed effects of meteorological conditions on the SO2 variations based on the differences of emissions and SO2 VCDs in South China. In general the scientific topic is meaningful, and the perspective of understanding effects of meteorology on SO2 depositions and dispersions is novel. However I have two major concerns below: 1) A credible emission inventory is quite a foundation of the study. However the authors do not give a peer-reviewed publication of the emission inventory in Section 2. The authors should cite some papers to introduce the methodology and validation of the

inventory. Meanwhile it could be more convincing if the authors do the same analysis based on another available peer-reviewed emission inventory. 2) The author did not consider any effects of regional transports in the discussion of the discrepancy of SO2 VCD and emissions. SO2 life time could be long and has a big variability. SO2 could be transported by on an order of 100 km, especially during night.

Specific Comments: 1) Line 116-117: how significant is the improvement of the new product on the study of variation trends? Do the authors compare the variation trends based on the new product with those based on the previous product? 2) Line 120-121: What kind of background correction is applied? Can the correction cause artifacts of some weak signals of SO2 in some regions which are dominated by the natural sources as discussed in Line 151-155? 3) Fig. 3a: The author should explain the line around 40 N latitude with high values in winter. 4) Line 161 and Fig. 2a: snow could cover the surface in the western and northern part of China in the seasons, except summer. The snow covered surface could impact the retrievals of SO2 VCD. This could be the reason of the missing values of satellite SO2 VCDs in the two regions, especially in winter. Do the authors consider the point in the discussion? Meanwhile in the lines of 161-163, the authors attribute the higher SO2 amounts in summer than other seasons to the natural emissions. However the snow coverage effect could also play a role. 5) Line 159: The authors conclude that "nearly half of the annual totals is released in winter" because of the significant higher SO2 VCD in winter than in other seasons. However SO2 lifetime could be also longer in winter. The larger SO2 VCD values could be also related to longer lifetime of SO2 due to its easy accumulations in winter.

---

## Author Comment (AC1) · 24 Oct 2018

We are grateful to reviewer #1 for the valuable and constructive comments that are helpful to improve the manuscript. We have tried to follow all of the suggestions, and make changes accordingly in the main text with red color.

*This paper uses OMI SO2 retrievals to study the effects of emissions and meteorology on SO2 loading over eastern China during 2005-2016. Monthly OMI SO2 from BIRA DOAS retrievals are compared with estimated SO2 emissions from the China*

[Figure]

*Statistical Yearbook. The authors show that OMI-observed SO2 has decreased significantly over eastern China, particularly for areas with the strongest emissions. They use an EOF analysis to demonstrate that the change is not monotonic and has four phases, with SO2 increasing during 2005-2007, decreasing sharply during 2007-2008 and 2014-2016, and only slightly increasing or decreasing during 2008-2013. They also show that the emissions and OMI SO2 are highly correlated over northern part of the country, but less so for southern China. They propose that abnormally dry and stagnant conditions over southern China during 2008-2013 may have caused a slight increase in SO2 loading, despite continued reduction in emissions. While several studies have examined the recent changes in SO2 pollution over China using satellite data, this study attempts to provide a somewhat different perspective. The conclusion that meteorology may play a fairly prominent role in the inter-annual changes in SO2 over southern China is interesting. The paper is well-organized and figures are mostly clear. However, I am not completely convinced that the emission data used can fully support the conclusions drawn in the study. I'd recommend that major changes be made before the paper can be accepted for publication in Atmos. Chem. Phys.*

**Reply:** First of all, we would like to again acknowledge reviewer for the positive comments on our paper. We agree that it is essential to assess the reliability of the emission data used in the study. We have addressed this issue through (i) verification of the sulphur emission statistics released in the official yearbook (OYB), and (ii) repetition of the analyses using another independent inventory.

To verify the OYB inventory and corroborate findings, the Multi-resolution Emission Inventory for China (MEIC) developed by Tsinghua University is adopted. In addition, two other estimates of national annual totals (REASv2 and Zhao) are also used. The references about MEIC, REASv2 and Zhao are listed in the main text.

Overall, the results based on MEIC confirm and reinforce the conclusions obtained

by OYB. Firstly, the comparison between the four databases show that, despite the spread in their magnitude, the reported temporal variations are characterized by a similar behavior. We conclude that this justifies the use of OYB as a main source of information. Please see details in Section 2.2 and Figure 1. Secondly, the main conclusions drawn from OYB are not modified when the MEIC dataset is used. This can be judged from the pairwise comparisons listed below and the associated interpretations in the main text.

There certainly exists non-trivial uncertainties related to the currently available emission inventories, especially when considering small spatial or temporal scales or when specific sectors are targeted. Accordingly, we have added a comment in the new Section 6 to highlight current issues and challenges that need to be addressed in the future.

**Table** pairwise comparison figures between OYB and MEIC

| **OYB** | Figure 2c | Figure 9a | Figure 10 | Figure 11 |
| --- | --- | --- | --- | --- |
| **MEIC** | Figure 2d | Figure 9b | Figure S3 | Figure S4 |

*Specific comments: The authors indicate that the emission data used in this study have very strong seasonal changes in SO2 emissions from China (almost half of emissions in winter, and only 10% in summer). But this is quite different from many previously published emission inventories which generally suggest a much smaller seasonal change (such as HTAP). Also according to a number of previous studies, the residential sector is in general estimated to contribute roughly 10% of all SO2 emissions. This is quite different from what is shown in Figure 9 of this study. The authors may consider using*
Interactive

comment

*a different, more widely recognized emission inventory for their analysis and check if their conclusion still stands.*

**Reply:** In the sentence "nearly half of the annual totals is released in winter when intensive heating takes place", the wordings "nearly half" and "released" are not appropriate leading to some confusion. In fact, Figure 3 actually depicts the seasonal variation of SO2 VCDs rather than SO2 emissions. In the revised version of the manuscript, we have reformulated the statement as follows: about 35% of the annual totals is taken up by winter, while SO2 in summer only accounts for 15%; the remaining 50 percent is almost equally divided in between spring and autumn. Please see Lines 228-231.

To explain the pronounced seasonal cycles in SO2 concentration, an additional figure (Figure 4) has been introduced which correlates the annual cycle of SO2 VCDs with sulphur emission, precipitable water and temperature at the four hotspots. Although intensive heating during winter in North China raises sulphur release, the variability of the SO2 emissions alone is not sufficient to drive the pronounced seasonality of SO2. The remaining variation is associated to seasonal changes in the meteorological conditions. The observed seasonality of atmospheric SO2 loadings is therefore resulting from variations of both emissions and meteorology. Further details are given in Section 3.2.

In Figure 9 (Figure 11 in the revised version), the Y-axis in a positive direction does not start at zero. We do this because our aim is relative change rather the absolute magnitude. If the origin of Y-Axis is set to zero, the blue bar denoting industrial emission is too tall to recognize the delta change.

*It also appears that the emission data used here are on a provincial level (and not*

*gridded) and the authors calculate the emission strength based on the area of each province. Can the authors confirm that? If so, how do the authors calculate total emissions (for example those in Figure 8) for a domain that partially covers several different provinces? Also note that the emissions and SO2 loading can be quite inhomogeneous even within the same province.*

**Reply:** We confirm that emission data used here are on a provincial level. Meanwhile, the terms "emission" or "emission amount" always refers to "emission strength", defined as emitted SO2 per unit area. To make it clear, we emphasize this in Lines 206-209.

Regional averaged quantities for North or South China are estimated as a weighted average by assigning the district area as a weight. Such explanation is added in Lines 334-336. In fact, the North China and South China are delimited by tracing provinces' boundaries, so that the North China and South China domains completely cover several different provinces. The demarcation line between the two portions are added in Figure 9.

We understand that emission and SO2 loading can be quite inhomogeneous within a same region, but this is not considered in this study due to the limitation that only continuous emission data on provinces are gathered at hand. Meanwhile, to match emission data, the gridded SO2 VCD is aggregated to provincial level. Consequently, the analysis given in Section 4 is constrained to provincial or multi-provincial levels. In Section 6, we particularly point out that future studies should use both gridded SO2 VCDs and gridded SO2 emission inventories. Please see Lines 458-462.

*It is not clear how the "north" and "south" are defined in this study. One would assume that Cheng-Yu, PRD, and YRD are all part of the "South". But the SO2 time series*

*in Figure 6 indicates that they have different trends during 2008-2013. How would the authors explain these different trends when Figure 10 appears to show generally similar meteorological conditions for the three regions?*

**Reply:** Thanks for your suggestion. We added a demarcation line in Figure 9 that separates South China and North China.

   We also realize that Figure 6 may not be clear enough to convey the major idea. This is because the monthly anomalies mask the primary signal, and trend is also not a suitable indicator for rebound phenomenon. Therefore we designed a new figure (Figure 8) which presents the temporal evolution of annual SO2 from 2005 to 2016 in each province of eastern China, with the segment over 2009-2013 highlighted by red color. It confirms that the SO2 does not evolve in a monotonic way but shows a striking rebound during 2009 to 2013. This pattern is observed throughout most of the region, with only two exceptions: the Guizhou and Guangdong provinces that experienced a monotonic decrease since 2005. See Figure 8 and detailed explanation in Lines 305-313.

*The "sudden downward shift of household emissions" in the south is quite surprising. Do the authors have an explanation for this? Or is this simply indicative of methodology change in the emission inventory?*

**Reply:** We have double checked and can confirm that the OYB inventory shows a sudden downward shift of household emissions for South China. Unfortunately, the official year book doesn't provide any clue whether this shift results from a change in methodology or reflects real changes in emission inventory. As a result, we decided to avoid any speculation in the main text.
We also note that the MEIC inventory does not shown evidences for such an "abrupt-shift" behavior. Nevertheless, both inventories do report a reduction of household emissions in South China, irrespective of the exact manner. The phrase "a sudden downward shift of household emissions" has been removed from the abstract and the conclusions. Instead we refer to "the coordinated cuts of industrial and household emissions". Please see Lines 39-40 and 438.

*Figure 4: there seem to be some negative SO2 values in the figure? Can the authors confirm that?*

**Reply:** In this study, all negative SO2 values are eliminated prior to the research. Thus, there are no negative values in Figure 4 (Figure 6 in the revised version). The label bar includes 'less than 0' bin, because we expect it to be symmetric. However, no patch on the shaded plot indicates less-than-0 value.

In addition, the YRD and PRD regions are slightly enlarged in the revised manuscript, to mitigate possible uncertainties.

*Figure 8: What is the unit for emissions? What does each data point represent in the scatter plot?*

**Reply:** The unit for emission is ton/km2, and each data point corresponds to one year and one province. Such key notes are inserted into the figure caption. Please see Figure 10.

*Figure 10: which area is the vertical profile in (d) for?*

**Reply:** The area is 23-31°N, 105-122°E rectangle. In addition, we find that the result is not sensitive to the chosen area.

*Writing: the authors should also make an attempt to improve the writing. Short, simple sentences in some cases may make the paper easier to follow.*

**Reply:** Thank you very much for your suggestion. Further polishing has been applied in the revised version.

---

## Author Comment (AC2) · 24 Oct 2018

We thank reviewer #2 for his suggestions to improve our manuscript. We have done our best to address each of them. The revisions in the manuscript are highlighted by red color.

*General comments: In this paper, Ting Wang et al. analyzed the spatial distribution and temporal variation trends of SO2 VCD and emissions in different regions of China in the last decade based on the OMI observation and emission inventory. Further they*

[Figure]

*discussed effects of meteorological conditions on the SO2 variations based on the differences of emissions and SO2 VCDs in South China. In general the scientific topic is meaningful, and the perspective of understanding effects of meteorology on SO2 depositions and dispersions is novel. However I have two major concerns below:*

**Reply:** We would like to again acknowledge the reviewer for his positive comments on our manuscript.

*1) A credible emission inventory is quite a foundation of the study. However the authors do not give a peer-reviewed publication of the emission inventory in Section 2. The authors should cite some papers to introduce the methodology and validation of the inventory. Meanwhile it could be more convincing if the authors do the same analysis based on another available peer-reviewed emission inventory*

**Reply:** Thanks for this valuable suggestion. The official estimate of SO2 emission inventory is published every year, and it is certain that peer review processes are undertaken before publication although these are not open to the public. The yearbook also briefly introduces the methodology to build the inventory, which has been added in Section 2.2. Unfortunately, no further detailed information is provided in the yearbook.

We understand the crucial concern regarding the reliability of the emission data used. To guarantee the robustness of this study, we have sought to address the concern in the revised version as you suggested. A similar comment was raised by the first referee and we refer to our corresponding reply.

*2) The author did not consider any effects of regional transports in the discussion of the discrepancy of SO2 VCD and emissions. SO2 life time could be long and has a big variability. SO2 could be transported by on an order of 100 km, especially during night.*

**Reply:** Thank you for this comment. The effect of regional transport is of importance, and climate-chemical coupled models are usually applied for its evaluation. Alternatively, we have used an Effective Emission Index (EEI) that accounts for the impact from both local and remote sources. Adopting results published by Zhang et al. (2015), we get: for North China, within-region SO2 emission contribute 68% followed by 19% from South China and 13% from other regions; for South China, within-region emissions provide 66%, while transport from North China and other regions amounts to 17% and 17% respectively. Based on these statistics and assuming that the EEI is linearly dependent on N and S and that external contributions remain fixed, the EEI index is formulated. For comparison purpose, we also define an Emission Index (EI) that considers single effect from within-in region emission. The detailed procedure to construct the indices are presented in Lines 339-365.

Note that we only consider the inter-regional transport between North China and South China. Because the spatial scale is large, we find that integrating the role of inter-regional transport does not alter the overall pattern and result. However, the best way to unveil fine scale details of transport is using climate-chemical coupled models. Due to the limited resources available for this study, this could unfortunately not be attempted. Future directions are highlighted in Section 6.

*Specific Comments:*
*1) Line 116-117: how significant is the improvement of the new product on the study*

*of variation trends? Do the authors compare the variation trends based on the new product with those based on the previous product?*

**Reply:** Compared to the BRD OMI NASA SO2 product, the BIRA retrievals proved to be better both in terms of noise level and accuracy. This product also includes a full characterization (errors, averaging kernels, etc.). The improved OMI PCA SO2 product of NASA show similar performance and long-term trends as the BIRA product. The BIRA SO2 product has also been validated in China using long-term MAX-DOAS data (Theys et al., 2015; Wang et al., 2017). As these comparisons have been done before this study, we added a short remark in Lines 125-130.

*2) Line 120-121: What kind of background correction is applied? Can the correction cause artifacts of some weak signals of SO2 in some regions which are dominated by the natural sources as discussed in Line 151-155?*

**Reply:** The correction we use here is based on a parameterization of the background values that are then subtracted from the measurements. The scheme first removes pixels with high SZA (>70°) and SCDs larger than 1.5 DU (measurements with presumably real SO2) and then calculates the offset correction by averaging the SO2 data on an ozone slant column grid. This is done independently for each across-track position and hemisphere, and the correction makes use of measurements averaged over a time period of two weeks around the measurement of interest. The details of background correction can be found in Theys et al. (2015).

Yes, the low level SO2 columns are subject to large uncertainties and the background correction is an important source of error. However, the regions with weak SO2 signals/background SO2 are not the subject of the present paper.

*3) Fig. 3a: The author should explain the line around 40 N latitude with high values in winter*

**Reply:** There is a belt of large positive values extending along 40°N in winter. However, it is a known artefact due retrieval limitations at large solar zenith angles. This does not incur any barrier to subsequent investigations, since our focus is eastern China that does not include this belt. Please see details in Lines 273-275.

*4) Line 161 and Fig. 2a: snow could cover the surface in the western and northern part of China in the seasons, except summer. The snow covered surface could impact the retrievals of SO2 VCD. This could be the reason of the missing values of satellite SO2 VCDs in the two regions, especially in winter. Do the authors consider the point in the discussion?*

**Reply:** Following your suggestion, we added Figure S2 and a paragraph to elaborate on this point. Figure S2 is designed to evaluate the availability of monthly SO2 data relative to the entire period. As mapped in Figure S2, there appears to be a substantial fraction of data gaps in western and northeastern China, especially in the winter half year. This can be attributed to snow cover surfaces and high solar zenith angles, which invalidates the measurability. However, the missing data issues in northeast and western China have virtually no impact on our study, because we mainly focus on the highly polluted eastern China.

Please see Figure S2 and relevant interpretations in Lines 135-145.

*Meanwhile in the lines of 161-163, the authors attribute the higher SO2 amounts in summer than other seasons to the natural emissions. However the snow coverage effect could also play a role.*

**Reply:** In the light of the considerable data gaps in western China as shown in Figure S2, it is impossible to draw firm conclusion. Therefore, we remove this statement in the revised version.

*5) Line 159: The authors conclude that "nearly half of the annual totals is released in winter" because of the significant higher SO2 VCD in winter than in other seasons. However SO2 lifetime could be also longer in winter. The larger SO2 VCD values could be also related to longer lifetime of SO2 due to its easy accumulations in winter.*

**Reply:** Thank you for this suggestion. The lifetime does take an important role in shaping the seasonality of SO2 VCD. To better explain the pronounced seasonal cycles in SO2 concentration, an additional figure (Figure 4) has been included to illustrate the annual cycle of SO2 VCDs and its relation to sulphur emission, precipitable water and temperature at the four hotspots. On the one hand, intensive heating during winter in North China raises sulphur release. However, emission alone is not adequate to explain the pronounced seasonality of SO2. Temperature and humidity are cold and dry in winter due to the influence of winter monsoon, which jointly weakens the rate of oxidation and wet deposition. Accordingly, it is expected that SO2 molecules will have a longer lifetime and will thus accumulate easier, as you suggested. The opposite is true for summer, when chemical reaction is active and wet removal is effective. Please see Figure 4 and relevant interpretations in Lines 233-243.

---

## Author Response (AR2)

**Technical Corrections**

We are very grateful to the editor for the timely feedback, and the reviewers for the further suggestions. Our responses can be found below, and the revisions in the manuscript are highlighted by red color. Meanwhile, all figures in vector format are packaged and uploaded as a separate document. Please kindly use these figures for production.

*Please check figure numbers throughout the text. For example, "Figure 3a" in line 268 should be "Figure 5a"?*

**Reply:** This mistake is now corrected and please see Line 270 and 273. Further, we have double checked all the citation of figures.

*Figures 2 and 3: I'm a bit puzzled by "tropospheric SO2" mentioned in figure captions – I assume those should be total column amounts?*

**Reply:** Yes, it is "$SO_2$ total columns" rather than "tropospheric $SO_2$". Such mistakes are corrected in the revised version, and please refer to Line 607 and 613.

*Figure 8: The Y-axis for the inserts needs to be marked – since different areas have different trends (as shown in Figure 5), I assume the range (in DU) for each insert is also different?*

**Reply:** Yes, the Y-axis range is different for each subplot. Following your suggestion, the Y-axis tickmarks are added. Please see the revised Figure 8.

*Also some further polishing would be helpful – for example in line 229, change "taken up by winter" to "from winter".*

**Reply:** The phrase "taken up by winter" is replaced with "from winter". Please see Line 234 and Line 425.

*For the reply to my second point in the specific comment, although you elaborate that "the low level SO2 columns are subject to large uncertainties and the background correction is an important source of error", you do not make any changes in Line 151-155 of the previous manuscript. One clarification needs to be added there.*

**Reply:** Such explanation is added in the revised version, and please see Lines 222-226.